# Tribological and Emission Behavior of Novel Friction Materials

**Ana Paula Gomes Nogueira** * , **Davide Carlevaris, Cinzia Menapace** and **Giovanni Straffelini**

Department of Industrial Engineering, University of Trento, 38123 Trento, Italy;
davide.carlevaris@alumni.unitn.it (D.C.); cinzia.menapace@unitn.it (C.M.); giovanni.straffelini@unitn.it (G.S.)
* Correspondence: ana.gomesnogueira@unitn.it

**Abstract:** The tribological behavior and the related airborne particles emission of three copper-free automotive friction materials are investigated. The tests were conducted using a pin-on-disc tribometer equipped with a specifically designed clean-enclosure chamber for the emission measurement. Particle number concentration from particle size 0.3 μm up to 10 μm and the mass of emitted particles between 1 μm to 10 μm were measured. Particular emphasis was given to the chemical composition of the bulk materials, the friction layers and the emissions, in order to explain the acting wear mechanisms, and the recorded emission of airborne particles. The results indicate that the recorded emissions do not correlate with the friction coefficient and the wear rates, since the wear mechanisms exert a different influence on the tribological and emission behavior of the materials under study.

**Keywords:** friction material; pin-on-disc testing; airborne particle emissions; friction layer

## 1. Introduction

The growing interest for environmentally friendly friction materials to be used for vehicular brake systems has sparked research efforts aiming at developing new material compositions, which might effectively respond both to performance and low pollution requirements. The obtainment of reliable relationships between the identified wear mechanisms and airborne particles emissions are extremely important in order to increase the efficiency and quality of brake materials development.

Copper has been a fundamental constituent of friction materials not exclusively because of its thermal properties [1] but also for its ability to form compact secondary plateaus [1,2], which contribute to a lower thermal fade and lower wear of the braking system, respectively. It is added to the friction materials either in fiber or powder form and concentrations up to 20% are typically used [3]. However, airborne particles related to brake pads wear pose as an environmental [1,4,5] and health hazard due to their metallic content, in particular due to their copper content, by prominently damaging the human respiratory system [6–11]. For this reason, the amount of copper allowed in friction materials is undergoing increasingly restrictive legislative regulations.

Friction materials formulations without copper have been developed over the latest years and the achievement of braking performances corresponding to copper-containing friction materials is driving several research projects around the world. Two commercial Cu-free and one Cu-containing friction materials were pin-on-disc tested by Lyu et al. [12]. Both Cu-free materials displayed a comparable friction coefficient and wear rate as the Cu-containing material. Barros et al. [13] described a positive effect on the friction coefficient given by the decrease of copper content. However, the copper-free materials presented high wear rates. As already mentioned, the wear resistance is associated to the formation of quite compact secondary plateaus. In this regard, barium sulphate (BaSO$_4$) has been reported as a promising copper substitute, since, in suitable concentrations, it helps improving the compactness of the secondary plateaus [14]. The addition of graphite particles is also proposed [15]

and has shown the ability to easily form the friction layer, including also iron oxides, mainly from the wearing out of the counterface disc.

The secondary plateaus are formed by the wear debris that starts to be compacted by the simultaneous action of contact pressure, sliding shear stress and friction heat [16]. The primary plateaus, mainly formed by metallic fibers and other tough components of the friction material, act as support and nucleation sites for the formation of the secondary ones. Primary and secondary plateaus together form the friction layer, defined as a third body created between the pad and disc surfaces. The tribological behavior of the system is directly linked to this layer, since the friction forces act through the real contact area, i.e., through the friction layers. It is reported [17,18] that from 15% up to 60% of the total area of the pad is covered by friction layer and its formation and disruption is a dynamic process during all sliding contact.

- Airborne brake emissions can be defined as the result of wear debris released with aerodynamic diameter lower than 10 μm by the tribological system. Some studies [19,20] proposed that a range of 35% to 55% of the total brake system wear becomes airborne particles. Garg et al. [21] estimated that almost 35% of brake pad mass loss is emitted as airborne particles.

- Several parameters could affect the emissions, aside from the friction material composition itself, with braking pressure, sliding velocity and temperature (which is strongly dependent on the aforementioned parameters) being reported as the most important ones, also as the tribological properties are concerned. Alemani et al. [10] identified a critical temperature between 165 °C and 190 °C, characterized by a significant increase in number of emitted ultrafine particles, while the coarse ones decrease. Nosko et al. [22] pointed out that the ultrafine emitted particles above 200 °C rises by several tens of percentages in terms of mass. Kukutschova et al. [23] suggested that the increase in the ultrafine fraction is linked to the degradation and burning-off of the phenolic resin, featuring a typical ignition temperature of about 300 °C. Based on the chemical composition of the emissions, several studies [8,24–26] demonstrated that airborne particles, originating from both pads and disc materials, including iron, come from the cast iron disc, and contribute to around 60% of the total mass emitted, for low-metallic brake systems [27]. Mosleh et al. [28] proposed that fine particles come exclusively from the disc, whereas the coarse ones originate from the friction material. On the other hand, Wahlström et al. [29] pin-on-disc (PoD) tested low-metallic and NAO braking pads and found Fe, Cu, Ti, Al, O and carbonaceous species as the main constituents of fine emissions, indicating the contribution of the friction materials. Alemani et al. [25] showed that disruption of the friction layer leads to the emission of particles characterized by a flake-like morphology. Previous studies [30,31] demonstrated a chemical composition correspondence between friction layer and emissions, also showing the contribution of the disc. However, all mechanisms involved in the generation of emission are not fully understood as of yet.

- Three testing set-ups are mainly used for friction materials, depending on the testing scale: full-scale car brake systems, dynamometric benches and laboratory-scale pin-on-disc (PoD) tribometers. PoD testing has been proven to be a fast and valuable method of assessing friction material properties, in particular when looking at comparative results [32]. Different investigations [12,33,34] have demonstrated that useful information regarding the emission behavior and its correlation with tribological properties come up from pin-on-disc laboratory tests. The obtainment, on a lab-scale level, of reliable relationships among the identified wear mechanism and the relevant airborne particle emissions would be extremely important in order to increase the efficiency and quality of brake material development.

- The present work aims at investigating the tribological behavior and the related emissions from three commercial Cu-free friction materials using pin-on-disc tribological testing. We focused our attention on the friction layer characterization and its connection with the tribological and emissions results. A thorough comparison among the chemical composition of bulk material, secondary

plateau and emissions is presented, as well as their relationships. Moreover, the composition of the secondary plateaus is analyzed at two different depths.

## 2. Experimental Methodology

### 2.1. Materials

Three commercial low-metallic friction materials were investigated in this study. They are copper-free materials, code-named "Cu-free/A", "Cu-free/Ba" and "Cu-free/Fe". The elemental compositions of the materials were obtained by energy dispersive X-ray spectroscopy (EDXS), and they are presented in Table 1. The density of Cu-free/A is 2.08 ± 0.10 g/cm$^3$ and its main components are iron fibers, zinc powder, graphite, vermiculite, tin sulfide and abrasives, such as magnesium and aluminum oxides. Cu-free/Ba has a density of 2.05 ± 0.05 g/cm$^3$ and basically contains the same ingredients of Cu-free/A, with the addition of barite. Cu-free/Fe, characterized by a density of 2.06 ± 0.06 g/cm$^3$, contains magnesium and aluminum oxides, tin sulfide, magnesium silicate, chromite and, most of all, a large amount of iron fibers.

**Table 1.** Elemental composition, in weight percentage, of the three investigated friction materials.

| Element (wt %) | Cu-Free/A | Cu-Free/Ba | Cu-Free/Fe |
|:---:|:---:|:---:|:---:|
| O | 22.3 ± 1.5 | 17.2 ± 0.5 | 24.4 ± 2.9 |
| Mg | 9.6 ± 0.8 | 6.2 ± 0.1 | 13.2 ± 1.8 |
| Al | 8.2 ± 0.7 | 6.3 ± 0.9 | 7.8 ± 1.3 |
| Si | 3.8 ± 0.6 | 2.8 ± 0.3 | 2.7 ± 0.4 |
| S | 8.7 ± 0.9 | 11.4 ± 0.7 | 5.1 ± 0.5 |
| Ca | 4.0 ± 0.1 | 3.0 ± 0.0 | 1.6 ± 1.1 |
| Cr | 2.5 ± 0.4 | 1.8 ± 0.1 | 2.4 ± 0.2 |
| Mn | 0.0 ± 0.0 | 0.0 ± 0.0 | 0.0 ± 0.0 |
| Fe | 18.6 ± 1.5 | 12.2 ± 2.6 | 34.9 ± 4.9 |
| Zn | 12.8 ± 1.7 | 8.7 ± 0.5 | 2.1 ± 0.2 |
| Sn | 9.5 ± 1.3 | 5.4 ± 0.1 | 5.9 ± 1.7 |
| Ba | 0.0 ± 0.0 | 25.1 ± 0.4 | 0.0 ± 0.0 |

Discs made by pearlitic grey cast iron with 235 HV10 hardness, density of 7.15 ± 0.02 g/cm$^3$ and diameter of 60 mm were used as counterface for the tribological tests, and were machined directly out of the brake disc of a European C-segment car. It is important to note that pearlitic grey cast iron is the standard material used for the vast majority of commercial cars.

### 2.2. Pin-on-Disc Tests and Emissions Measurements

A horizontal pin-on-disc tribometer equipped with a specifically designed clean-enclosure for particle emission measurements was used for the investigation of tribological properties and related emissions. The dry sliding PoD tests were carried out at room temperature, with an applied pressure of 1 MPa and sliding velocity of 1.51 m/s, which correspond to mild wear, typical of standard braking conditions, and thus representative of common driving situations. Each test had a duration of 5400 s and it was preceded by a bedding test, conducted under the same conditions for 1800 s, to provide a conformal contact between the pin and the disc surfaces and to remove any irregularity from the mating surfaces.

Three pin specimens, extracted from real brake pads, with a diameter of 10 mm and height of around 16 mm were tested for each friction material composition. The dynamic friction coefficient, μ, was continuously recorded during the tests. An analytical balance with a sensitivity of $10^{-4}$ g was used to weight the pin and disc before and after the PoD tests. From these values, the wear coefficient ($K_a$) of the pins and of the whole system (pin + disc) was calculated based on the Archard relationship [35,36]:

$$Ka = V/Fs$$

where V is the wear volume of the system, F is the normal load applied and s is the sliding distance. The volume loss of the system was calculated as the sum of the volume losses of the disc and of the pin.

Figure 1 shows the overall test apparatus used, which is based on the approach developed by Olofsson et al. [33,34]. The fan (B) takes the ambient air (A) that passes through a High-Efficiency Particulate Air (HEPA) filter (C), which removes the particles providing a clean air entering inside the chamber (D). The feeding air velocity was set at 11.5 m/s and, considering the volume of the enclosed chamber, the used flow rate provided an air exchange rate of 99 times/h. Before every test, the cleanliness of the air flow inside the chamber was verified, showing a background concentration of airborne particles lower than 10 #/cm$^3$ for all the studied cases. The relative humidity and temperature were measured during each test, obtaining values in the range of 20–30% and 23–25 °C, respectively.

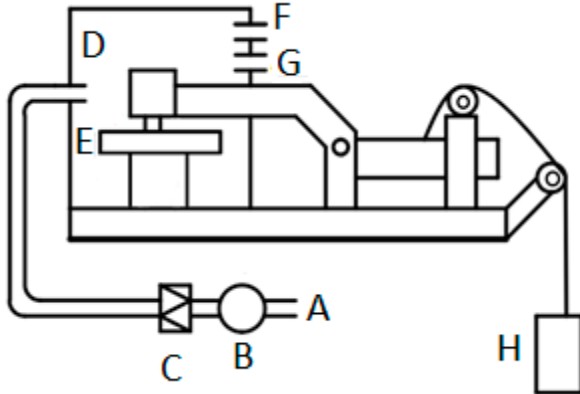

**Figure 1.** Schematic of test equipment. Ambient air (**A**), fan (**B**), HEPA filter (**C**), air inside the chamber (**D**), rotating disc sample (**E**), air outlet to Impactor PM10 (**F**), air outlet to the OPS (**G**) and weight (**H**).

A TSI® (TSI Incorporated, Shoreview, USA) Optical Particle Sizer (OPS), model 3330 and a Dekati® (Dekati Ltd, Tykkitie, Finland) PM10 Impactor were used in order to measure the number concentration, which is the number of particles per unit volume of air, and the mass of airborne emitted particles, respectively. Both instruments were connected simultaneously to the enclosed chamber during the PoD tests, in the position (G) for OPS and (F) for Impactor (Figure 1). For the mass measurements, the impactor was set up with an inlet flow rate of 9.7 L/min and the particles were collected on aluminum foils, divided into three different size ranges: >10 μm, from 10 μm to 2.5 μm and from 2.5 μm to 1 μm. These substrates were weighted before and after two tests for each friction material using an analytical balance with 10$^{-4}$ g sensitivity. The OPS measured the particle number concentration with size ranging from 0.3 μm up to 10 μm, divided into 16 channels, with a sampling frequency of 1 Hz. The instrument is able to measure a concentration range from 0 to 3000 particles/cm$^3$ and works with a self-controlled sampling flow rate of 1 L/min.

*2.3. Characterization Procedures*

The worn surfaces and transverse cross sections of the pins were analyzed using a JEOL® IT300 (JEOL Ltd., Tokyo, Japan) scanning electron microscope (SEM) equipped with a Bruker® EDXS system (Bruker, Billerica, MA, USA), having an XFlash 630M detector linked to it. In particular, SEM images of the friction layer were taken using both secondary and backscattered electrons to allow a better observation of the morphology of the entities. The same equipment was used to characterize the particles collected by the Impactor.

## 3. Results

### 3.1. Friction and Wear Behavior

Figure 2 shows the evolution with time of the friction coefficient from representative tests of each friction material. The three materials show an initial running-in period of approximately 1000 s, with a noticeable increase in the friction coefficient. Table 2 shows the average friction coefficient in the steady state (in the 3500 to 5400 s interval) for the three investigated friction materials. The results indicate that Cu-free/Fe is the material characterized by the highest friction coefficient (0.52), while Cu-free/A and Cu-free/Ba present similar values (0.43 and 0.40, respectively).

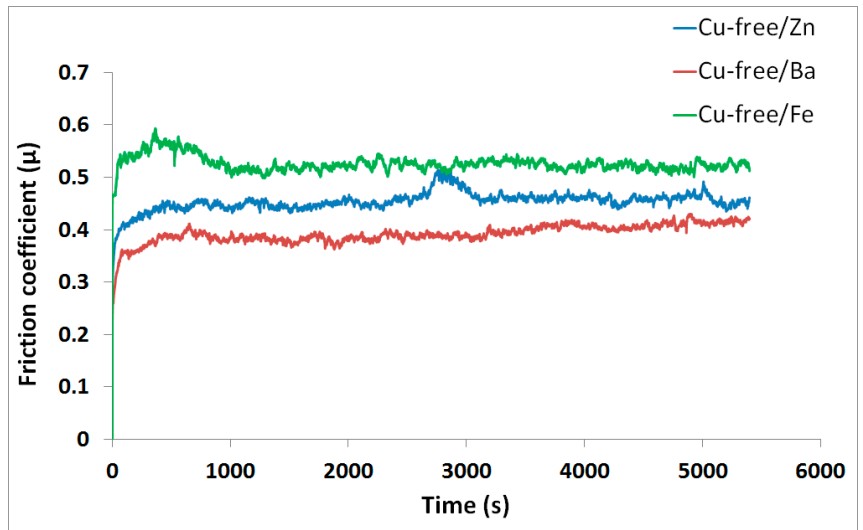

**Figure 2.** Friction coefficient evolution obtained from representative PoD tests conducted on the three friction materials.

**Table 2.** Average values of friction coefficient ($\mu$) at steady-state and experimental specific wear coefficients ($K_a$).

| Material | Friction Coefficient | Ka Pin m2/N) | Ka System (m2/N) |
|:---:|:---:|:---:|:---:|
| **Cu-free/A** | 0.43 ± 0.04 | $4.2 \times 10^{-14} \pm 0.92 \times 10^{-14}$ | $5.6 \times 10^{-14} \pm 1.29 \times 10^{-14}$ |
| **Cu-free/Ba** | 0.40 ± 0.02 | $5.4 \times 10^{-14} \pm 0.35 \times 10^{-14}$ | $6.9 \times 10^{-14} \pm 0.47 \times 10^{-14}$ |
| **Cu-free/Fe** | 0.52 ± 0.01 | $4.3 \times 10^{-14} \pm 0.06 \times 10^{-14}$ | $5.2 \times 10^{-14} \pm 0.07 \times 10^{-14}$ |

In Table 2, the experimental values of the specific wear coefficients, $K_a$, of the pin only and for the whole system are also included. The system specific wear coefficient takes into account the contributions of both the friction material and the counterface disc. The Cu-free/Ba material shows the highest specific wear coefficient, while Cu-free/A and Cu-free/Fe display close ones; however, Cu-free/A presents more scattered values.

### 3.2. Airborne Particles Emissions

Figure 3 shows the evolution with time of the total particle concentration from representative tests of each friction material (the same ones of Figure 2). The experimental curves exhibit a relatively high data scattering. However, it is possible to notice that Cu-free/Fe displays a concentration peak at around 500 s (see arrow), in correspondence of the running-in period observable in the friction coefficient shown in Figure 2.

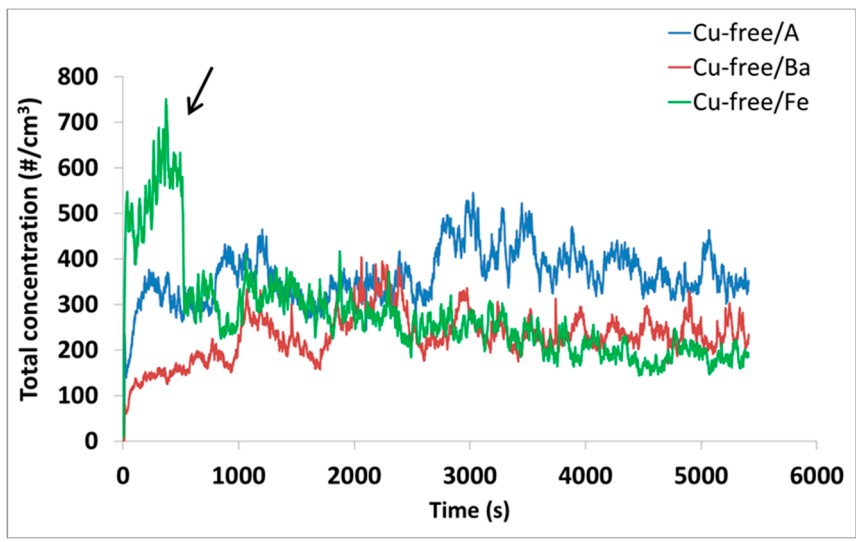

**Figure 3.** Particle concentration evolutions obtained from representative PoD tests conducted on the three friction materials. The arrow indicates a peak at around 500 s for the Cu-free/Fe material.

Figure 4 shows the total mass of collected particles and the average of the total particle concentration for each tested friction material. The data of total mass collected represents the total cumulative value measured for three samples of each material. The average concentration of airborne particles was calculated from the mean values of the three samples for each material, in the time interval between 1000s and the end of the tests. Regarding the mass, the Cu-free/A shows the highest collected amount, while Cu-free/Ba is characterized by the lowest value. It is hereby appreciable how Cu-free/A also shows the highest number concentration of emitted airborne particles, whereas Cu-free/Fe and Cu-free/Ba both average a value close to 230 #/cm$^3$.

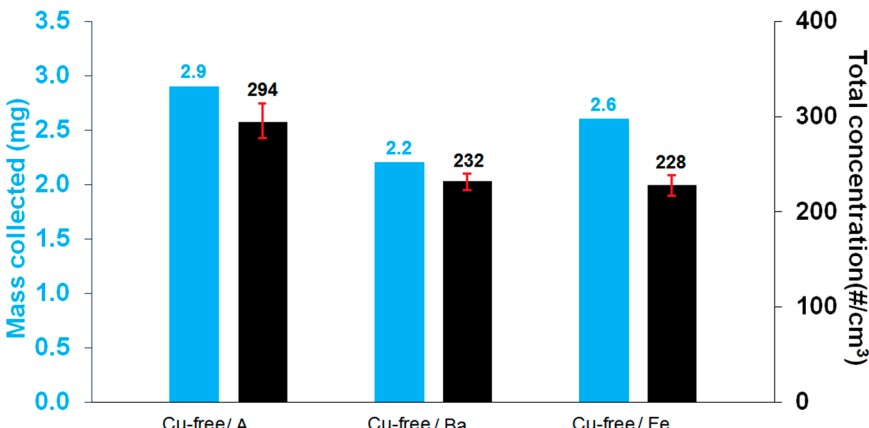

**Figure 4.** Mass of particles collected and average of number concentration of airborne particles at steady-state for the three tested materials.

### 3.3. Characterization of the Friction Layer and Emitted Particles

Figure 5 shows the worn surfaces of the pins at low magnification, as obtained from SEM analysis using backscattered electrons. A higher amount of reinforcing fibers (brighter regions) is visible in the Cu-free/Fe material, in accordance to the data from the compositional Table 1. SEM images at higher magnifications allow to better evaluate the contact plateaus, as can be seen in Figure 6. The figures at the top were obtained using backscattered electrons, whereas the corresponding SEM images at the bottom were obtained using secondary electrons. The presence of the secondary plateaus close

to iron fibers that act as primary plateaus can be clearly appreciated. It is also possible to see that the secondary plateaus in Cu-free/Fe are much more compacted than the secondary plateaus in the other two materials, appearing darker and more uniform on the secondary electrons image, therefore indicating better sintering of the debris. The secondary plateaus in the Cu-free/Ba material seems to be the less compacted ones, and the Cu-free/A contain some small cracks.

The elemental compositions of the secondary plateaus are presented in Table 3, as obtained from EDXS analysis. The same elements found in the virgin materials (Table 1) are all present in the secondary plateaus, but in different concentrations. In particular, an increased amount of iron is observed in the three materials, indicating the contribution from the counterface disc wear.

Figure 7 shows SEM backscattered electrons observations of the cross section of the worn pins, where the secondary plateau can be distinguished (red highlighting). The Cu-free/Fe material displays a more compact secondary plateau cross section than Cu-free/A and Cu-free/Ba, in agreement with the worn surface observations (see Figure 6).

Table 4 shows the elemental compositions of the cross-sectional portions of the secondary plateaus for the three tested materials, which appear to be in accordance to the worn surface compositions show in Table 3. However, the oxygen presents a lower content.

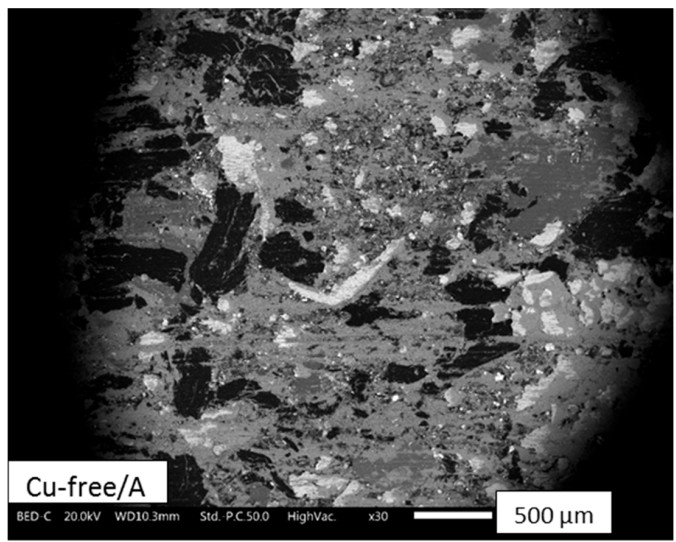

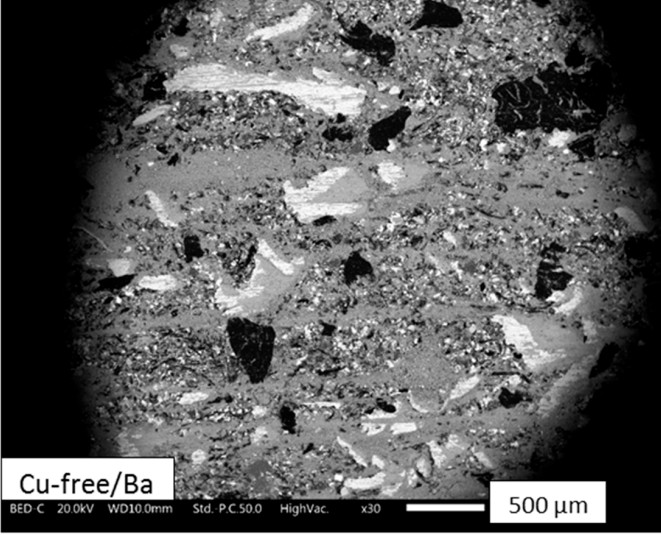

**Figure 5.** *Cont.*

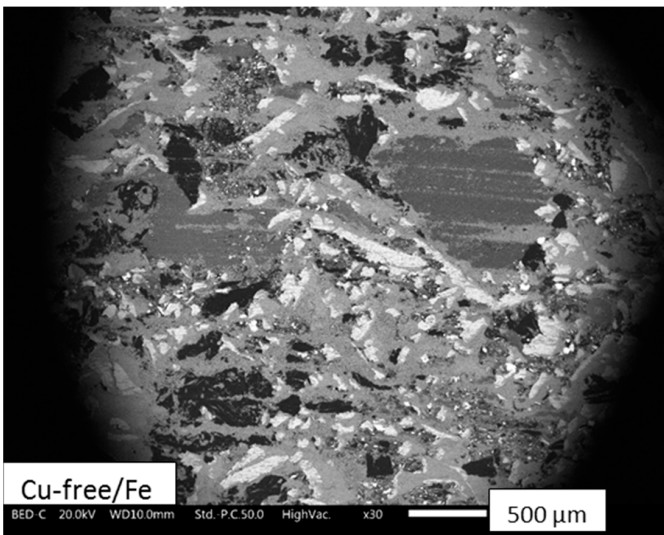

**Figure 5.** SEM backscattered electrons observations of worn pins' surface.

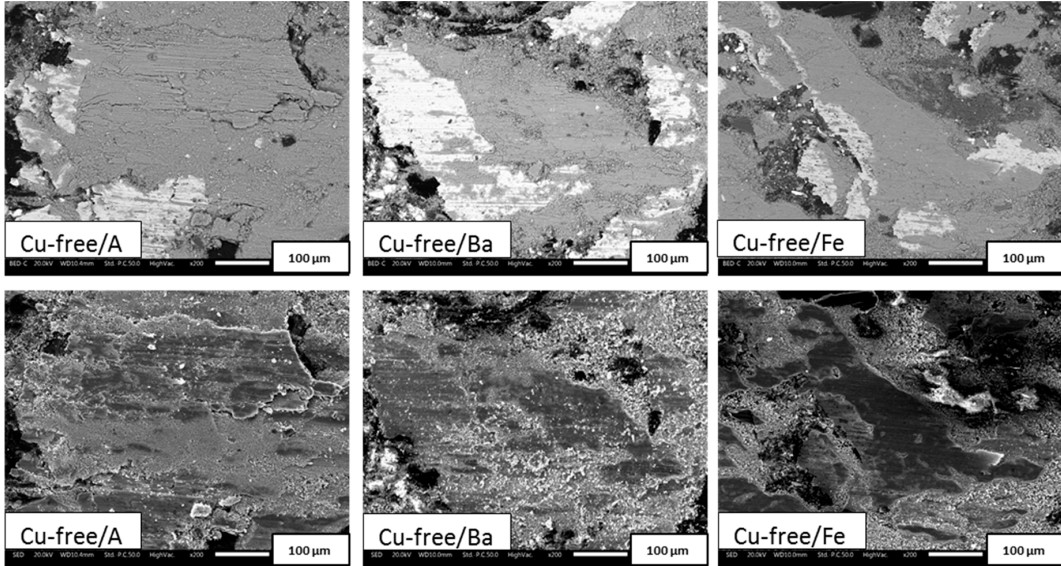

**Figure 6.** SEM high magnification observations of worn pins' surface. Above: backscattered electrons images. Below: secondary electrons images.

**Table 3.** Elemental compositions, in weight percentage, of the secondary plateau from worn surface of the three friction materials.

| Element (wt %) | Cu-Free/A | Cu-Free/Ba | Cu-Free/Fe |
|:---:|:---:|:---:|:---:|
| O | 12.6 ± 1.5 | 13.3 ± 0.4 | 12.7 ± 0.6 |
| Mg | 1.8 ± 0.3 | 1.3 ± 0.1 | 1.9 ± 0.1 |
| Al | 1.9 ± 0.1 | 1.3 ± 0.1 | 1.2 ± 0.2 |
| Si | 1.4 ± 0.1 | 1.3 ± 0.2 | 1.4 ± 0.1 |
| S | 2.2 ± 0.0 | 3.7 ± 0.4 | 1.5 ± 0.1 |
| Ca | 0.6 ± 0.1 | 0.5 ± 0.1 | 0.5 ± 0.1 |
| Cr | 1.0 ± 0.0 | 0.9 ± 0.2 | 0.7 ± 0.2 |
| Mn | 0.3 ± 0.0 | 0.3 ± 0.0 | 0.0 ± 0.0 |
| Fe | 71.6 ± 3.0 | 64.6 ± 1.7 | 76.9 ± 1.3 |
| Zn | 3.8 ± 0.2 | 2.8 ± 0.5 | 0.4 ± 0.0 |
| Sn | 2.6 ± 0.2 | 2.2 ± 0.1 | 2.7 ± 0.1 |
| Ba | 0.0 ± 0.0 | 8.0 ± 0.3 | 0.0 ± 0.0 |

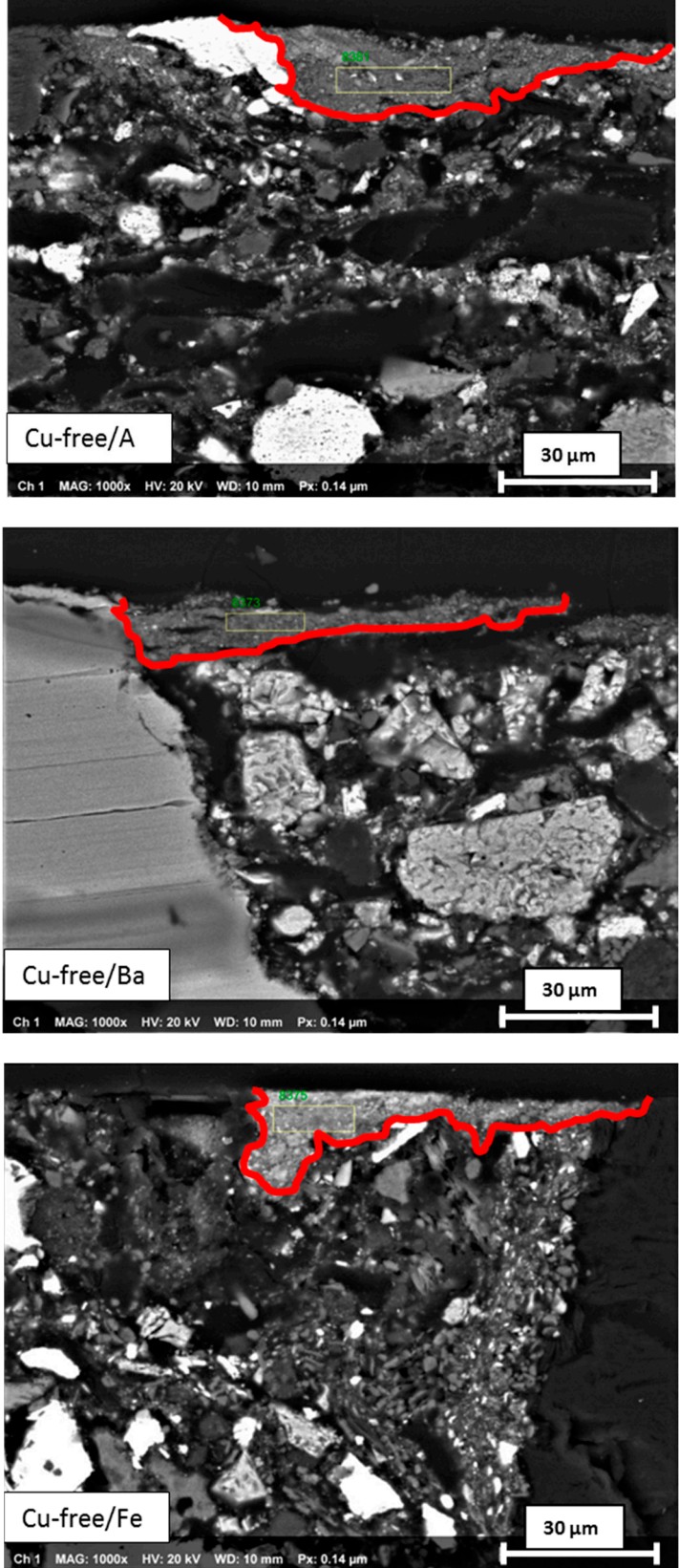

**Figure 7.** SEM observations of the pins' cross section. Secondary plateau is highlighted in red.

**Table 4.** Elemental composition, in weight percentage, of the secondary plateau from cross section of the three friction materials.

| Element (wt %) | Cu-Free/A | Cu-Free/Ba | Cu-Free/Fe |
|:---:|:---:|:---:|:---:|
| O | 9.1 ± 0.5 | 4.2 ± 1.1 | 4.9 ± 0.1 |
| Mg | 1.3 ± 0.3 | 2.6 ± 0.3 | 2.4 ± 1.0 |
| Al | 0.8 ± 0.2 | 2.3 ± 0.3 | 3.7 ± 0.2 |
| Si | 0.5 ± 0.1 | 3.2 ± 1.6 | 3.0 ± 1.4 |
| P | 0.0 ± 0.0 | 0.0 ± 0.0 | 0.7 ± 1.0 |
| S | 3.8 ± 0.3 | 5.1 ± 0.6 | 2.5 ± 0.1 |
| Ca | 1.2 ± 0.1 | 0.8 ± 0.1 | 0.8 ± 0.1 |
| Cr | 0.8 ± 0.1 | 0.5 ± 0.1 | 1.6 ± 0.4 |
| Mn | 0.4 ± 0.0 | 0.2 ± 0.1 | 0.0 ± 0.0 |
| Fe | 72.9 ± 1.5 | 65.2 ± 1.7 | 75.5 ± 0.5 |
| Zn | 5.9 ± 0.5 | 4.2 ± 0.7 | 0.0 ± 0.0 |
| Sn | 3.4 ± 0.7 | 2.6 ± 0.6 | 4.9 ± 0.3 |
| Ba | 0.0 ± 0.0 | 9.1 ± 0.8 | 0.0 ± 0.0 |

Figure 8 shows the SEM backscattered electrons micrographs of airborne particulate emissions for the three friction materials, collected by the impactor filter of the 2.5–10 μm aerodynamic diameter range. The three materials show the presence of both coarse flake-shaped debris and smaller rounded particles. The elemental composition of these airborne particles is shown in Table 5, acquired through EDXS analysis. The compositions of the friction layer (Tables 3 and 4) and the emitted particles (Table 5) are very similar.

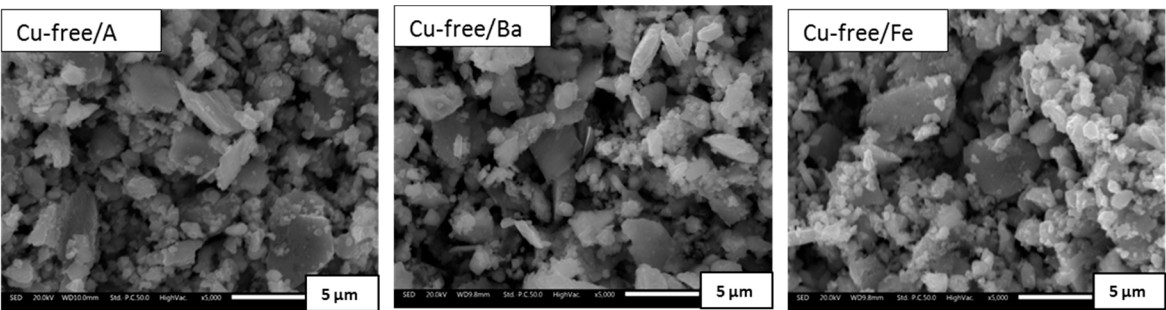

**Figure 8.** SEM micrographs of the airborne particles in the10–2.5 μm range of the three friction materials.

**Table 5.** Elemental composition, in weight percentage, of the airborne particles in the 10–2.5 μm range of the three friction materials.

| Element (wt %) | Cu-Free/A | Cu-Free/Ba | Cu-Free/Fe |
|:---:|:---:|:---:|:---:|
| O | 7.1 ± 1.6 | 11.8 ± 0.7 | 9.3 ± 1.1 |
| Mg | 1.9 ± 0.2 | 1.9 ± 0.1 | 1.6 ± 0.5 |
| Al | 1.7 ± 0.1 | 1.9 ± 0.1 | 1.6 ± 0.7 |
| Si | 1.7 ± 0.1 | 1.5 ± 0.1 | 0.5 ± 0.2 |
| P | 0.0 ± 0.0 | 0.0 ± 0.0 | 0.0 ± 0.0 |
| S | 1.7 ± 0.1 | 3.7 ± 0.2 | 2.5 ± 0.2 |
| Ca | 0.0 ± 0.0 | 0.3 ± 0.1 | 0.0 ± 0.0 |
| Cr | 0.7 ± 0.0 | 0.7 ± 0.0 | 1.3 ± 0.1 |
| Mn | 0.4 ± 0.0 | 0.3 ± 0.1 | 0.2 ± 0.2 |
| Fe | 79.1 ± 1.7 | 64.5 ± 0.4 | 78.6 ± 5.1 |
| Zn | 3.0 ± 0.1 | 2.7 ± 0.0 | 0.0 ± 0.0 |
| Sn | 2.6 ± 0.3 | 2.5 ± 0.4 | 4.5 ± 0.2 |
| Ba | 0.0 ± 0.0 | 8.2 ± 0.2 | 0.0 ± 0.0 |

## 4. Discussion

The tribological and the associated emission behavior of friction materials are related to the characteristics of the friction layers that form on the pin surfaces during PoD sliding tests [25,31,37]. To discuss these aspects, it is useful to compare the compositions of the materials, the friction layers and the airborne particles, as shown in Figure 9a–c.

As already remarked, the iron content in the friction layers (Tables 3 and 4) is higher than in the bulk of the pins (Table 1) for the three materials, owing to the contribution from the counterface cast iron disc wear [16,17,25]. Even in Cu-free/Fe, which presents the lowest difference between the iron content values in the friction layers compared to the bulk material, the content of iron is twice higher in the contact plateaus. This would also explain the strong decrease of the other ingredients in the friction layers. In Figure 9, the content of some selected elements is reported, in order to highlight this decrease in content of some pad-exclusive compounds, such as $Al_2O_3$, MgO and $BaSO_4$, in passing from the bulk friction material to the plateaus and emissions. The iron content does not appear to vary between the external surface of the secondary plateaus (as revealed from the EDXS measurement on the top surface; it is indicated with 'plateau' in the Figure 9) and the internal part of the plateaus (referred to as 'cross section'). The oxygen percentage does instead change depending on the distance from the pin surface. In particular, it goes down when moving from the top surface to the inner part of the friction layer (for example, from 12.7 to 4.9 in case of Cu-free/Fe). Furthermore, the oxygen-containing compounds, such as $Al_2O_3$, MgO and $BaSO_4$, are usually present in higher amount in the plateaus cross section with respect to its surface (see Figure 9). All these remarks suggest that the observed oxygen gradient is mainly due to the oxidation of metallic iron present close to the surrounding environment, with the formation of iron oxides, such as magnetite and hematite [16,25], on the outer side of the friction layer.

Cu-free/Fe displays a comparatively higher iron content (34.9) with respect to the other friction materials (18.6 and 12.2); this is most likely due to the higher presence of steel fibers that promote an increased contribution of adhesive wear in the sliding against the cast iron counterface. In fact, the friction coefficient value (Table 2) and iron content in the secondary plateaus surface (Table 3) displays a correlation. Indeed, an increase in iron presence provides a higher tribological compatibility with the cast iron counterface [38], thus increasing the adhesive interaction between the mating surfaces during braking and the relevant adhesive contribution to friction. The friction coefficient values of the tested materials are in agreement with the other studies [39]; in fact, superior values were obtained for the materials characterized by a higher iron content.

The wear of the system (Table 2) may be linked to the compactness of the friction layers, which can be qualitatively defined as their degree of densification and structural soundness. Material Cu-free/Fe presents a well-compacted secondary plateau (see Figure 6 Third Column), and a low and stable wear coefficient ($5.2 \times 10^{-14}$). Instead, Cu-free/Ba shows the less compacted secondary plateau (see Figure 6 Second Column) and the highest wear ($6.9 \times 10^{-14}$). The compactness of the friction layers appears to be dependent on the iron in the secondary plateaus (Table 3). As seen, Fe is oxidized and under the action of the friction shear stresses and the frictional temperature rise, the oxides are sintered together to form compact glazes [16,40]. At the same time, the higher amount of Fe fibers in the Cu-free/Fe material facilitates the formation of more extensive plateaus (see bright regions in Figure 5) with respect to the other tested materials, thus contributing to the achievement of the discussed material performances. To summarize, in Cu-free/Fe a Fe-rich and well compact friction layer is formed, and this favored higher friction and lower wear. On the contrary, in Cu-free/Ba Fe in the friction layer was low, thus inducing a lower friction coefficient and a higher wear.

Focusing on the EDXS composition of the airborne particles (Table 5), it is possible to obtain information on their formation mechanisms by comparing the obtained results. By first considering the Cu-free/A friction material, the iron content in the produced particles (79.1) results are higher than the one recorded in both the inner (72.9) and superficial portions (71.6) of the secondary plateaus (see Figure 9a). Considering the corresponding trend of decrease in oxygen content (down to 7.1 from 9.1 and 12.6), it suggests that the airborne particles contain more metallic iron than the secondary

plateaus, especially when compared to the superficial secondary plateau. This in turn indicates that, for the Cu-free/A friction material, the airborne particles originate both from the disruption of the secondary plateaus as well as the direct wear of the cast iron counterface. This latter mechanism in particular is promoted by the abrasive wear exerted by the abrasives that are present in this friction material. This behavior can also be correlated with the higher emission, in terms of both number and mass (see Figure 4), exhibited by this material. For the other two friction materials, Cu-free/Ba and Cu-free/Fe, the iron content in the airborne particles is approximately similar to its content in the friction layer (Figure 9b,c), while the oxygen content is intermediate between the one of the external and the internal part of the plateaus, but skewed towards the latter. This entails that most of the airborne particles are formed from the disruption of the friction layer, and in particular of its external parts. It is also important to reiterate that the iron content is strongly lower in the starting material composition due to the fact that it has not yet interacted with the cast iron disc.

In general, the mass emissions are reported to be proportional to the system wear. However, for the materials under study there is no direct correlation between airborne emissions and wear. Therefore, wear mechanisms play a fundamental role not only in the amount of produced particles but also in their size, thus determining the wear debris fraction that will become airborne. This might explain the comparatively higher specific wear coefficient recorded by Cu-free/Ba respect to the other tested materials (Table 2), despite it showcasing the lowest emissions (Figure 4). Cu-free/Ba is characterized by not well compacted secondary plateaus, due to the low content of iron, in particular iron oxides, in the secondary plateaus. The comparatively lower emission of airborne particles presented by Cu-free/Ba can then be ascribed to the dynamic rupture of the friction layer into a higher fraction of relatively large fragments, which do not become airborne after abandoning the mating surfaces. This mechanism seems to be facilitated by barite particles, which provide a solid lubricant effect, reducing the adhesion of the plateaus onto the substrate.

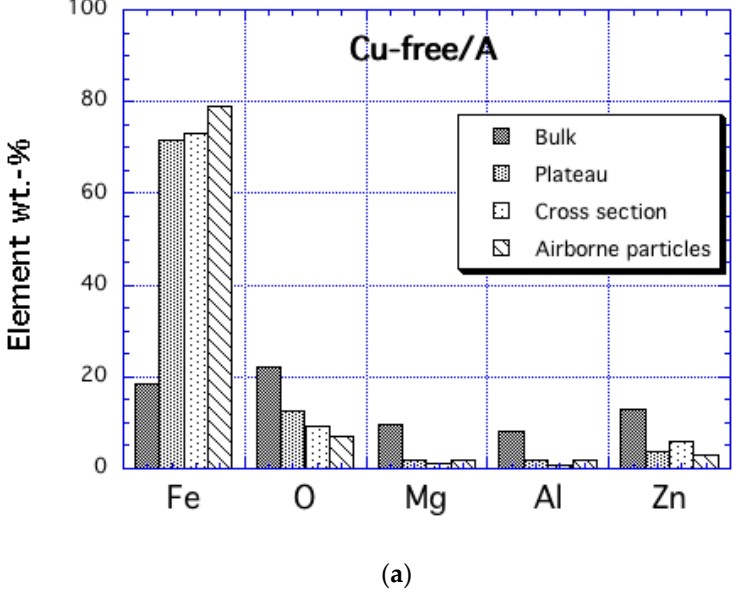

(**a**)

**Figure 9.** *Cont.*

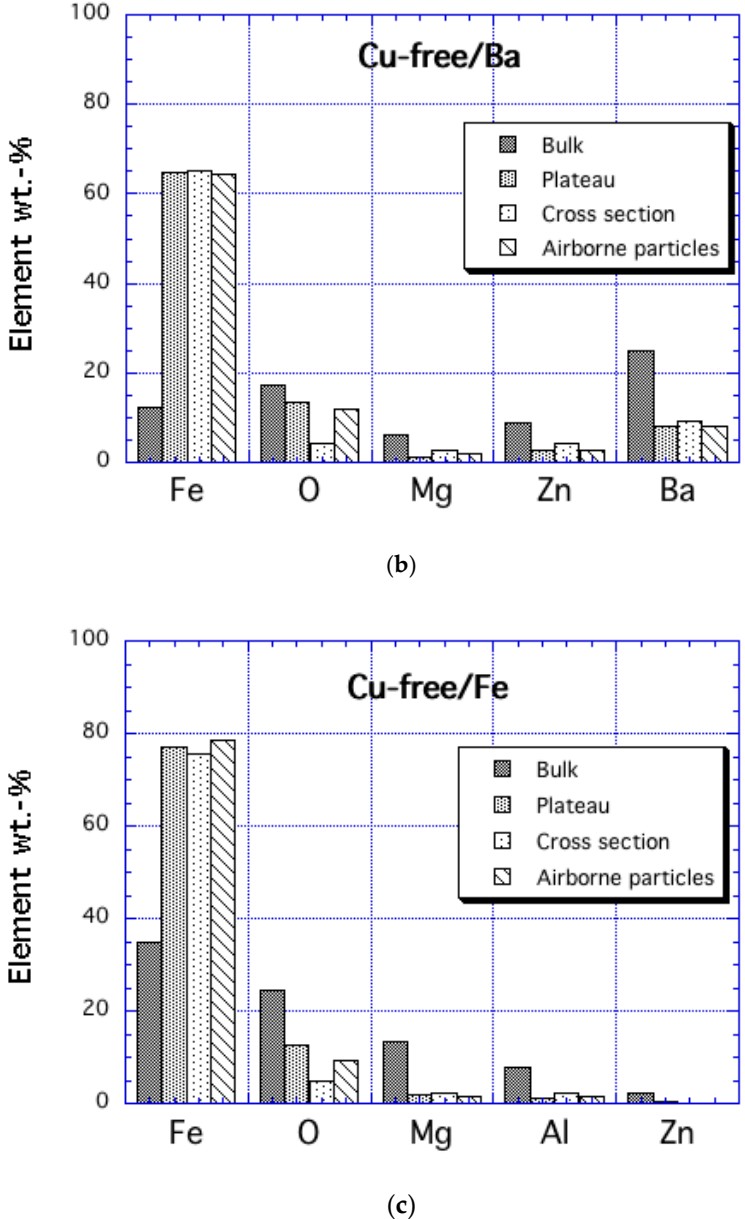

**Figure 9.** Comparison of the elemental composition found on bulk material, secondary plateau on the pin surface (plateau), secondary plateau on the pin cross section (cross section) and airborne particles for (**a**) Cu-free/A, (**b**) Cu-free/Ba and (**c**) Cu-free/Fe friction materials.

## 5. Conclusions

A comparative study was conducted on three copper-free friction materials in order to evaluate their tribological and particulate matter emission behavior. All materials underwent testing in a PoD tribometer, sliding against a grey cast iron disc counterface. By comparing the obtained results and previous research, the following conclusions can be drawn:

- The material Cu-free/Fe yields the higher friction coefficient and lower wear coefficient in comparison to the other two materials. Both results are linked to the higher amount of iron oxide present in the secondary plateaus, favoring the friction performance by the iron oxide contact with the cast iron counterface disc, thus decreasing the wear by the formation of more compact secondary plateau.

- The emissions produced by Cu-free/A friction material contain more metallic iron than its secondary plateaus. This indicates that the airborne particles are made not only by the disruption of the secondary plateaus but also by the cast iron counterface wear, exerted by the abrasives present in this friction material. For the other two friction materials, Cu-free/Ba and Cu-free/Fe, the airborne particles seem to be mainly formed by the disruption of friction layer, since both materials present a clear correspondence between the chemical composition of the emissions and the secondary plateaus.

- The material Cu-free/Ba exhibits a higher wear coefficient and lower emissions with respect to the other tested materials. The absence of correlation between the wear and emissions is associated to the disruption of friction layer in relatively large fragments caused by the not well compacted secondary plateau.

This study helped in obtaining some insights regarding how the friction material compositions influence the formation mechanisms and properties of the contact plateaus and, consequently, of the airborne emissions. This new information should be exploited to formulate future material compositions and they also give some directions on how to further study the discussed phenomena and entities.

**Author Contributions:** The contribution of A.P.G.N. was in terms of conceptualization, data curation, investigation, methodology and writing of the original draft. The contribution of D.C. was in terms of conceptualization, data curation, investigation and writing of the original draft. The contribution of C.M. was in terms of methodology and reviewing the writing. The contribution of G.S. was in terms of conceptualization, investigation, supervision, writing the original draft and reviewing the writing. All authors have read and agreed to the published version of the manuscript.

**Funding:** This research received no external funding.

**Acknowledgments:** The authors wish to thank Stefano Gialanella for helpful suggestions and discussions.

**Conflicts of Interest:** The authors declare no conflict of interest.

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
