# Peer review of "Tribological and Emission Behavior of Novel Friction Materials"

_atmosphere, doi:10.3390/atmos11101050_

Round 1

Reviewer 1 Report

This study investigates the tribological and airborne particle emission behavior of three Cu-free brake pad friction materials using a pin-on-disc tribometer located in a close clean chamber. Optical particle counter is used to measure the number concentration of emitted particles, which are also collected with filters for SEM+EDS observations. I suggest a minor revision before publishing the manuscript, in terms of taking the following comments.

Some small comments:

  1. Page 1 line 39. “helps the compactness of the secondary plateaus” is suggested to be “improving the compactness of the secondary plateaus”.
  2. Page 2 line 62. It should be “Chemical composition of the emissions”.
  3. Page 2 line 68. Should be “as the main constituents”.
  4. Page 2 line 73. “Friction behavior” should be “tribological behavior”, which includes friction and wear.
  5. I didn’t see any information about the density of the three friction materials and the grey cast iron brake rotor. Since the described test precedure only measured the mass loss of the samples so that the volume loss can only be calculated with corresponding density, in such a way demonstrating the specific wear rate of the system “pin+disc”. This information is important due to the different constitutions of the three brake pad friction materials.
  6. Page 4 line 147. It should be Table 2 shows the average CoF in the steady state. The same for line 149.
  7. I would like to ask that what does the standard deviation of the CoF, wear rate and average particle concentration look like for each material? As I interpret, the authors have done three repeats for each friction material whilst unfortunately the results only show the mean value but not the standard deviation.
  8. Page 5 last sentence. Should be “Friction coefficient shown in Figure 2.”

Reviewer 2 Report

This manuscript investigated the tribology performance and airborne particle emissions of three copper-free friction materials emphasizing on the role of friction layers in these two items. Some interesting results and observations are presented. However, some issues should be considered before it is accepted.

  1. Three pin specimens were tested for each friction material. How about the repeatability of the test? What is the scatter?
  2. Line 118, the background airborne particle concentration was measured before every test. It should be shown in the manuscript to indicate the error of the measurement.
  3. Line 148, the average friction coefficient in the steady-state stage for Cu-free/A is shown in Table 2. However, for Cu-free/Fe and Cu-free/Ba, the average friction coefficients at the end of the tests are shown. Why are they different?

Reviewer 3 Report

Review of Atmosphere manuscript (927836) entitled “Tribological and emission behavior of novel friction materials”, Nogueira et al..

This paper presents the results of three Cu-free materials friction tests with regard to tribology, particular matter, and debris/surface characteristics. In general, the research concept and experimental works are very well summarized, but repeatability tests and in-depth scientific analyses are expected to be improved for meeting the journal quality. The following comments could help to further improve the quality of the paper.

1. Data quality (reliability)
The most critical concern of this research is data reliability. This kind of researches (especially emissions and coefficient tests) must be repeated at least three times and observe any noticeable differences with error ranges. I hope the authors already performed repeatability tests, but this manuscript does not contain any repeatability test data and comments. It could be a bit frustrating for the authors, but this needs to be modified immediately for the publication. The SEM image is not mandatory, but please address error ranges in your compositions and error-bars in all of your figures. Or, future readers might not be able to solely trust your data.

2. Comments.
(1) Table 1 & bold consistency in all Tables
- Recommend changing the sulfur (S) to bold for the second column Cu-free/Ba. Or, change the naming “Ba” to “BaSO4” or so. It could be confused with pure Barium.
(2) Figure 2 description
- The arrow part for the Cu-free/Fe shows not a remarkable difference. That is why readers might want to see error ranges in this type of study. As you can see, the 3000 seconds’ Cu-free-A case shows a more noticeable peak than the arrow. Also, there is not enough analysis regarding the reasons even in the discussion part.
(3) Figure 3 Add more experimental data
- PM data is very sensitive to environments even like small temperature, humidity, HVAC conditions, and so on. Without any repeatability test results, removing particle concentration results would be better.
(3) Figure 4 Add error bars
- Same. Without any error bars, the OPS and Dekati impactor results might not be worthy to show.
(4) Discussions
- Almost all descriptions are very qualitative. All manuscripts would be improved by including quantitative metrics for each of the qualitative statements, such as “Ba exhibits a higher wear coefficient and lower emissions” to “Ba exhibits “XXX times” higher wear coefficient and “YYY times” lower emissions “compared to Fe or A”…”.

3. Summary
- If the authors do not address enough reliability statements and in-depth analyses more, this could be a technical report, not a research paper. The manuscript shows very enough potential to be improved, so I expect a better version of the revised manuscript.

Reviewer 4 Report

This study investigated three commercial Cu-free friction material and their friction behavior and associated emissions when applied in vehicular brake systems. The topic is interesting and of value considering the environmental impact of such systems. And authors demonstrated a well-designed experimental set-up, using a pin-on-disc tribometer, for the purpose of this evaluation and presented very interesting results. However, there are aspects that have not been addressed very well in the manuscript.

General Comments:

1)The merit and context of the work is not well-established. The necessity of the work and its contribution to this area of research is not clear. It is not discussed how the outcome of this study could improve the environmental impacts of brake systems. Conclusion section is mainly a summary of result and discussion sections, but authors could use this section to discuss the implication of their work; if any of the discussed material, or any specific component, or any composition mix is found to be environmentally superior; what do authors suggest to be done after this experiment which could help put these findings into the context; etc. Also, authors could add a few sentences in introduction section about the studied health effects (maybe include some numbers on these effects) to show and emphasize the importance of the work.

2) The experiment set-up is explained in details, but it is not discussed what other testing methods are available and why authors selected pin-on-disc tribological testing for this evaluation.

3) Authors stated in the introduction that temperature has been reported as the most important parameter to affect the emissions, but this parameter is not accounted for in this study, neither as a separate nor influential one.

4) Is there any previous study which results from this study could be compared with? For example, same methodology is reported to be used by one of the cited papers, Wahlström et al. Do their findings (if relevant) support the finding by this study?

5) Is there any reason for the material selection in this study? Why these three? Are they the most common ones? Can they represent all/most of the materials being used in braking systems? Are they currently competing for the market?

Specific Comments:

1) Line 69: How does the study by Alemani et al. fit in this paragraph? Is "a flake-shape emission particle" considered fine PM?

2) Line 90: Provide reason for this selection or add discussion on how the results might change with other specifications.

3) Line 93: Table 1

Add more explanations to the caption and provide more details for the "BOLD" numbers. This is relevant to Table 3. This has not been done in Table 4 and Table 5. Any reason not?

4) Line 99: Could the results be translated for the case of “nonstandard” brakes? If so, how? Is there any sensitivity analysis conducted?

5) Line 121: Figure 1

Use larger font size for the "letters".

6) Line 124: What is “the number concentration”? Please define.

7) Line 149: Please explain why it is legit to consider only the end of the tests. If it is legit, why was this applied only for two of the material?

8) Line 154: Figure 2

Authors may add to the caption regarding the arrow. Figure should be self-explanatory and capable to stand alone by itself.

9) Line 160: Table 2

Update table title to be consistent:

  • Table 2. Average values of friction coefficient (μ) at steady-state and experimental specific wear coefficients (Ka).
  • Add a note to the table, state the fact that averaging times are different for the first two items.

10) Line 169: Figure 3

Add a caption explaining the arrow.

11) Line 173: Add argument on why three samples are representative. Discuss how comprehensive these three are of the whole period. Were the samples collected at the same time-steps for all three material. If not, how could that affect the comparison?

12) Line 243: An all-in-one table merging Table-1, Table-3, Table-4 and Table-5 could be helpful for easier comparison.

13) Line 275: Is there any measurement or definition of "compactness"? Not clear how Figures 6 and 7 supports this statement or serve as demonstration of “compactness”.

14) Line 317: Figure 9

It is a not clear how does the Iron share (i.e., percentage) increase in composition of surfaces and airborne particles? If the percentage of Fe increases on the surface, it should mean that higher share of the other contents should be found in the airborne particles. Please add more explanations on this outcome.

Round 2

Reviewer 3 Report

All of the first round comments were applied in the revised manuscript and addressed enough in the response. 

Reviewer 4 Report

The edits and additional information that authors provided to address the comments are appreciated and the work is in a good shape to be published in Atmosphere. Still, I believe that the merit and context of the work could be better established and importance of the results is an important poorly discussed part in this work.